# Aspiration Pneumonia After ERCP Under Anesthesiologist-Administered Sedation: Prevalence, Risk Factors and Clinical Outcomes of an Underestimated Adverse Event

**DOI:** 10.3390/medicina61122172

**Published:** 2025-12-06

**Authors:** Nicolò de Pretis, Emilia Calderini, Silvia Maria Mora, Camilla Cerioli, Maria Vittoria Galli, Maria Cristina Conti Bellocchi, Armando Gabbrielli, Katia Donadello, Gianluca Brazzo, Luigi Martinelli, William Mantovani, Luca Frulloni, Stefano Francesco Crinò

**Affiliations:** 1Gastroenterology and Digestive Endoscopy Unit, University of Verona, 37124 Verona, Italy; 2Gastroenterology and Digestive Endoscopy Unit, University of Trento, 38122 Trento, Italy; 3Department of Surgery, Dentistry, Gynaecology and Paediatrics, University of Verona, 37124 Verona, Italy; 4Anesthesia and Intensive Care Unit B, University Hospital Integrated Trust of Verona, 37124 Verona, Italy; 5Clinical Epidemiology Unit, Azienda Provinciale Servizi Sanitari, 38123 Trento, Italy

**Keywords:** ERCP, adverse event, pneumonia, aspiration pneumonia, complication

## Abstract

*Background and Objectives*: Aspiration pneumonia is a well-described complication of upper digestive endoscopy. However, limited data are available on incidence, risk factors and clinical consequences of post-endoscopic retrograde cholangiopancreatography (ERCP) aspiration pneumonia (pEP). *Materials and Methods*: All consecutive ERCPs performed under anesthesiologist-administered sedation at the Endoscopy Unit of the University of Verona between 1 April 2022 and 31 August 2024 were retrospectively evaluated. Demographic, clinical and endoscopic data were collected. *Results*: One thousand one hundred forty consecutive ERCPs were included. The main indication was malignant biliary stricture, and the patient’s mean age was 68 ± 13.9 years. Overall incidence of pEP was 2.7%. The American Society of Anesthesiologists (ASA) score, presence of active cholangitis before ERCP and performance of endoscopic ultrasound (EUS) and ERCP in the same sedation session were significantly associated with a higher risk of pEP at both univariable and multivariable analysis. pEP was an independent risk factor for post-ERCP 30-day mortality and for prolonged hospital stay. *Conclusions*: pEP is a relatively frequent adverse event after ERCP. In patients with a high ASA-score, active cholangitis and scheduled EUS and ERCP in the same sedation session, preventive medical and/or anesthesiological strategies might be considered. Additional prospective studies are needed to confirm these data.

## 1. Introduction

Endoscopic retrograde cholangiopancreatography (ERCP) is a relevant technique in the management of pancreatobiliary disorders. Since it is an invasive procedure, it is associated with complications that range from mild to severe, with an overall reported mortality up to 0.33–1% [1,2].

The most commonly reported and studied ERCP-related adverse events are acute pancreatitis (3.5–9.7%), bleeding (0.3–9.6%) and duodenal perforation (0.08–0.6%) [3]. ERCP is a complex and unpleasant procedure that must be conducted under sedation or general anesthesia. Therefore, the ERCP potential risks should also include those related to sedation. Limited data are available regarding sedation-related adverse events in patients undergoing ERCP. Few and anecdotal cases of aspiration pneumonia have been reported, despite it being a well-documented adverse event in upper endoscopy [4,5,6] with potentially fatal consequences. Christensen et al. reported an incidence of post-ERCP aspiration pneumonia (pEP) of 0.93% among 1177 prospectively enrolled ERCP patients. However, a clear definition of pEP was lacking, and risk factors and clinical consequences were not investigated [2]. Similar data have been published in a study from the US, including 528 prospective ERCPs, performed mainly in the prone position and under anesthesiologist-administered sedation. The authors reported pEP in 0.38%, but the study was mostly focused on intra-procedural sedation-related adverse events, and no specific analysis was performed on risk factors and clinical consequences of aspiration pneumonia [6].

In a recent study from South Korea on 162 patients undergoing ERCP in the prone position and conscious sedation for acute biliary pancreatitis with cholangitis, pEP was described in 8 patients (4.9%) with a significantly higher risk (Odds Ratio of 4) if the procedure was performed within 18 h from the clinical onset of acute pancreatitis [7]. Furthermore, a retrospective study from Japan [5], including 287 consecutive ERCPs under conscious sedation in patients with native papilla (no data were available on patient position), reported pEP in 1.05%, with a significantly higher risk in patients with Eastern Cooperative Oncology Group performance status > 2.

A German prospective registry on acute sedation-associated complications in gastrointestinal endoscopy analyzed 368,206 endoscopies, reporting a 0.01% rate of major complications and 0.3% of minor complications. Identified risk factors included an American Society of Anesthesiologists (ASA) score > 2, gastroscopy, ERCP and emergency-therapeutic procedure. However, aspiration pneumonia was not investigated because it was not considered an acute complication [8].

Other smaller studies focused on sedation-related adverse events during ERCP without including pEP [1,9,10].

Definitive data on prevalence and risk factors of pEP are lacking. The identification of patient-related and ERCP-related risk factors for pEP may be useful in clinical practice to select which patients should undergo orotracheal intubation.

The main aim of this study was to investigate the incidence of pEP in a tertiary academic center. Our secondary aims were the identification of potential risk factors for pEP and investigating the impact of pEP on 30-day mortality and hospital length of stay (LOS).

## 2. Materials and Methods

All consecutive ERCPs performed at the Endoscopy Unit of the University of Verona were retrospectively evaluated on a prospectively maintained database between 1 April 2022 and 31 August 2024. The study was approved by the local ethical committee (3369CESC), and written informed consent was obtained from all patients for the procedure following the Declaration of Helsinki.

At our institution, all patients undergoing ERCP are hospitalized for at least 2 days after the procedure. Rectal Indometacin is routinely administered.

ERCPs are performed under an anesthesiologist-administered sedation during spontaneous breathing and in the supine position. General anesthesia and tracheal intubation are rarely performed for ERCP, according to anesthesiologists’ preference. Patients receive anxiolysis (midazolam, 1 mg iv), light opioid analgesia (fentanyl, 50–100 mcg iv) and undergo the procedure using Marsh or Schnider target-controlled propofol infusion models (according to anesthesiologist’s preference), aiming at Richmod Agitation-Sedation (RASS) Scale of −2/−3. When deeper sedation plans are obtained, advanced airway management is always required and applied.

Fasting for at least 8 h before the endoscopic procedure is required. According to the internal organization of the Endoscopy Unit, EUS and ERCP can be performed in the same anesthesia session only if a free double slot in the endoscopy schedule is available. Otherwise, the scheduling of EUS and ERCP is separate.

A chest X-ray performed within 5 days before the procedure is mandatory for pre-ERCP anesthesiologic evaluation. Patients undergo clinical and biochemical observation during the 48 h after ERCP and are discharged if no complications arise. If complications arise or if the clinical conditions deteriorate, hospitalization is extended.

Demographic, clinical and endoscopic data were collected. Measured outcomes were the incidence of pEP, post-endoscopic hospital LOS and overall 30-day mortality.

pEP diagnosis was considered in case of new-onset fever (>37.5 °C) and/or cough arising within 48 h after ERCP, with new-onset radiological evidence of pneumonia on chest X-Ray performed within 72 h from ERCP.

Length of hospital stay was calculated starting from the day of ERCP until the day of discharge.

Exclusion criteria were the following:-lack of pre-ERCP chest X-ray;-pre-ERCP radiological diagnosis of pneumonia;-age < 18 years;-ERCP performed under general anesthesia and tracheal intubation.

### Statistical Analysis

Continuous variables were summarized using mean and standard deviation (sd) or median and quartiles (p25–p75), according to variable distribution, and compared between groups using the Wilcoxon rank-sum test. Categorical variables were reported as frequencies and percentages and compared using the chi-square test.

Univariate analysis was performed to evaluate which factors were associated with post-ERCP pneumonia and 30-day mortality. The relationship between categorical predictors and outcomes was assessed using chi-square tests, while continuous variables were analyzed using Wilcoxon rank-sum tests. Additionally, Spearman’s rank correlation coefficient was used to assess the association between the ASA-score and both pneumonia and mortality outcomes.

Negative binomial regression models were used to analyze predictors of hospital length of stay.

For multivariable analysis, odds ratios were calculated using Firth logistic regression to assess independent predictors of post-ERCP pneumonia and 30-day mortality, given the potential issue deriving from the modest number of pneumonia and death events. The model for pEP included as covariates age, sex, malignancy, ASA-score, cholangitis, double procedure, and procedural complications. The model for 30-day mortality included pneumonia, age, sex, ASA-score, malignancy, and procedural complications. Hospital LOS was modeled using a negative binomial regression, incorporating pneumonia, age, sex, ASA-score, malignancy, and procedural complications as covariates.

Multicollinearity was assessed using variance inflation factors (VIFs). The final multivariable models were selected based on clinical relevance and statistical significance. All analyses were conducted using Stata software v.16.1.

## 3. Results

One thousand one hundred eighty-two ERCPs were performed at our institution during the study period. Forty-one patients were excluded because ERCP was performed under general anesthesia and tracheal intubation. Additionally, one more patient was excluded because the pre-ERCP X-ray was lacking. Therefore, 1140 ERCP procedures were included in the analysis. Among these, 653 patients were males (57.3%), and 487 (42.7%) were females, with a mean age of 67.5 ± 14.0 years. Clinically significant comorbidities were present in 589 patients (51.7%), with a median ASA-score of 2 (p25–p75: 1–2).

The most frequent indication for ERCP was malignant biliary stricture (50.1%), followed by biliary stones (23.7%) and benign biliary stricture (15.6%). Moreover, in 87 patients (7.6%), active cholangitis was diagnosed before ERCP.

The overall cannulation rate was 92.4%. Cannulation was considered difficult in 278 patients (24.4%) and standard in 861 patients (75.6%). Finally, in 107 patients (9.4%), ERCP was performed immediately after endoscopic ultrasound during the same anesthesiologist-administered sedation. Main demographic, clinical and endoscopic data are reported in Table 1.

### 3.1. Post-ERCP Aspiration Pneumonia (pEP)

pEP was diagnosed in 31 patients (2.7%). The incidence of pEP was higher in patients with at least one clinically significant comorbidity (3.7% vs. 1.6%; *p* = 0.029). Moreover, the incidence of pEP was significantly and progressively higher as the ASA-score increased, ranging from 0.6% in the ASA-score for one patient to 10.0% in the ASA-score for four patients. Moreover, at univariable analysis, the incidence of pEP was significantly higher in patients with cholangitis (6.9% vs. 2.4% in patients without; *p* = 0.013). Finally, in patients who were submitted to both EUS and ERCP during the same anesthesiologist-administered sedation, the incidence of pEP was higher, even if this difference did not reach statistical significance (5.6% vs. 2.4%; *p* = 0.054). Univariate analysis is reported in Table 2.

In the multivariable analysis, a higher ASA-score, the presence of active pre-ERCP cholangitis, and the performance of EUS and ERCP under the same anesthesiologist-administered sedation were identified as independent risk factors for an increased incidence of pEP (Table 3).

### 3.2. 30-Day Mortality and Hospital Length of Stay

The overall 30-day mortality was 2.5% with 28 deaths among the 1140 patients included in the study. Mortality was significantly higher in patients with malignant biliary stricture compared to patients with benign disease as an indication for ERCP (4.4% vs. 0.5%, respectively; *p* < 0.001). At univariable analysis, we found that higher 30-day mortality was also associated with cardiological comorbidity (5.0% vs. 1.4%; *p* < 0.001), age (*p* = 0.028) and ASA-score (*p* = 0.003). Univariable analysis is reported in Appendix A. In multivariable analysis, malignant biliary stricture (OR 7.54 CI 2.35–24.16), pPE (OR 5.24 CI 1.53–17.98), female sex (OR 2.43 CI 1.09–5.40) and ASA-score (OR 1.97 CI 1.10–3.51) were significantly associated with a higher 30-day mortality risk.

Finally, the median hospital LOS was significantly longer in patients who developed pEP (26.5 ± 21.5 days) compared to patients who did not develop pEP (9.6 ± 12.0 days), *p* < 0.001. In multivariable analysis, malignant biliary stricture (IRR 1.28, CI 1.16–1.40), pPE (IRR 2.42, CI 1.86–3.14), and ASA-score (IRR 1.15, CI 1.08–1.23) were significantly associated with longer hospital stay. Non-difficult cannulation (IRR 0.68, CI 0.62–0.76) and female sex (IRR 0.83, CI 0.76–0.91) were significantly associated with shorter hospital LOS. Univariable analysis on hospital LOS is reported in Appendix A.

Multivariable analysis for 30-day mortality and length of hospital stay is reported in Table 4.

## 4. Discussion

To the best of our knowledge, this is the first study focusing on the risk of pEP in patients undergoing ERCP with anesthesiologist-administered sedation. Our results suggest that pEP is relatively common (2.7%) and might have significant consequences in terms of hospital LOS and 30-day mortality. The most frequent indication for ERCP was malignant biliary stricture, which is not surprising considering that the Endoscopy Unit of the University of Verona is a tertiary center for pancreatic and biliary diseases. Despite that, the included population well represent all different possible indications to ERCP and data on cannulation achievement and difficult cannulation are in line with published literature [11,12,13].

As expected, the ASA-score, which assesses the overall patient health, was significantly associated with a higher risk of pEP incidence. Our data are in line with those published by Berzin et al., showing that the ASA-score is associated with an increased risk of intra-procedural sedation-related events during ERCP under sedation and endotracheal intubation [6].

Interestingly, age did not appear to be associated with pEP at adjusted and unadjusted analyses, suggesting that the risk might be more influenced by comorbidity than age. These data confirm the hypothesis that ERCP is not associated with a significantly higher complication rate if performed in older patients, as suggested by recent studies from both China and Japan [14,15].

An additional risk factor for pEP at adjusted and unadjusted analyses was active pre-ERCP cholangitis. Cholangitis is a well-recognised risk factor for ERCP-related adverse events, such as bleeding, probably because the septic process contributes to weakening the clinical status [16,17]. Therefore, in patients undergoing ERCP, the presence of active cholangitis should be considered together with the ASA-score to estimate the overall patient’s clinical status and related risks. These data are similar to those of Lee and colleagues [7], reporting the risk of post-ERCP aspiration pneumonia up to 4% in patients with acute biliary pancreatitis and cholangitis, supporting the hypothesis that cholangitis might represent a susceptibility factor for pEP.

Interestingly, performing EUS and ERCP within the same sedation session was clearly associated with an increased risk of pEP in both unadjusted and adjusted analyses. Possible explanations are the prolonged sedation time, the double esophageal intubation, and the change in patient’s position from lateral for EUS to supine for ERCP, which may lead to aspiration of pharyngeal secretions. A recent paper published by our group [18] comparing the performance of EUS and ERCP within the same versus in separate sessions in patients with malignant distal biliary strictures demonstrated no differences in terms of cannulation rate and safety. However, the number of included patients was limited, aspiration pneumonia was not considered as a single and specific adverse event, only malignant distal biliary strictures were analyzed, and the recruitment period was different compared to the present study.

Our data might suggest that in patients with an ASA-score > 2, cholangitis or scheduled EUS and ERCP in the same sedation session (or more than 1 of these factors), preventive strategies might be considered to reduce the risk of pEP. No clear and definitive medical or anesthesiological strategies have been identified to avoid aspiration pneumonia. Oro-tracheal intubation and early post-procedure chest X-ray with possible initiation of antibiotic therapy might be considered and investigated.

In the present study, pEP appeared to be a clinically relevant event, considering the significant association with a higher 30-day mortality rate and prolonged hospital LOS. Therefore, the identification of risk factors for pEP and the development of specific preventive strategies might have a relevant clinical impact in patient management. 

This study’s strengths are represented by its original focus on the risk for pEP, the large sample size of more than 1100 consecutive ERCPs and the use of rigid diagnostic criteria for pEP diagnosis (symptoms and chest X-ray within 72 h after ERCP) together with the constant performance of pre-ERCP chest X-ray, limiting the risk of overestimation.

We acknowledge some study limitations: first, the retrospective design, despite partially compensated by the prospective maintained database; second, the analysis was performed on patients undergoing ERCP exclusively in supine position which might limit the possibility of extending these data to ERCPs performed in prone or left-side position; finally, considering that the Endoscopy Unit of the University of Verona is a reference center for pancreatic diseases, a significant proportion of the included cases was represented by patients with malignant biliary stricture.

Therefore, additional prospective and multicenter studies are needed to confirm these data and to investigate the role of prone position and orotracheal intubation in pEP development.

## 5. Conclusions

In conclusion, pEP is a relevant adverse event of ERCP performed under anesthesiologist-administered sedation, impacting mortality and length of hospital stay. An elevated ASA-score, cholangitis and the same sedation session EUS and ERCP appear to be significant risk factors for pEP. Preventive medical and anesthesiological strategies might be considered in selected patients.

## Figures and Tables

**Table 1 medicina-61-02172-t001:** General characteristics of the included population.

Included Patients, n (%)	1140 (100)
Female, n (%)	487 (42.7)
Age-years, mean (SD)	67.5 (14.0)
Comorbidity, n (%)	589 (51.7)
- Diabetes, n (%)	220 (19.3)
- Cardiovascular, n (%)	299 (26.3)
- Pulmonary, n (%)	95 (8.3)
ASA-Score, median (p25–p75)	2 (1–2)
Indication for ERCP	
- Malignant stricture, n (%)	571 (50.1)
- Biliary Stones, n (%)	270 (23.7)
- Benign stricture, n (%)	
- Other, n (%)	
Cholangitis, n (%)	87 (7.6)
Previous sphincterotomy, n (%)	477 (41.9)
Cannulation, n (%)	1053 (92.4)
Difficult cannulation, n (%)	278 (24.4)
- Double wire, n (%)	92 (8.1)
- SEPTOTOMY, n (%)	32 (2.8)
- Precut, n (%)	27 (2.4)
EUS + ERCP, n (%)	107 (9.4)

Abbreviations: N, number; SD, standard deviation; p25–p75, first-third quartile; ASA-score, American Society of Anesthesiologists score; EUS + ERCP, execution of both endoscopic ultrasound (EUS) and endoscopic retrograde cholangiopancreatography (ERCP) during the same anesthesiologist-administered sedation session.

**Table 2 medicina-61-02172-t002:** Unadjusted analysis on the incidence of post-ERCP aspiration pneumonia (pEP).

	Patients, n (%)	pEP, n (%)	*p*-Value
Included Patients	1140 (100)	31 (2.72)	/
Sex			0.929
- Females	487 (42.7)	13 (2.67)	
- Males	653 (57.3)	18 (2.76)	
Age group			
- <60 years	279 (24.5)	6 (2.2)	0.394
- 60–69 years	316 (27.7)	8 (2.5)	
- 70–79 years	334 (29.3)	10 (3.0)	
- ≥80 years	211 (18.5)	7 (3.3)	
Indication for ERCP			
- Malignant stricture	572 (50.2)	16 (2.8)	0.637
- Biliary Stones	270 (23.7)	5 (1.9)	
- Benign stricture	170 (14.9)	5 (2.8)	
- Other	128 (11.2)	5 (4.1)	
Comorbidity			
- Yes	589 (51.7)	22 (3.7)	0.029
- No	551 (48.3)	9 (1.6)	
Diabetes			
- Yes	220 (19.6)	11 (5.0)	0.021
- No	920 (80.4)	20 (2.2)	
Cardiovascular			
- Yes	299 (26.2)	15 (5.0)	0.005
- No	840 (73.8)	16 (1.9)	
Pulmonary			
- Yes	95 (8.3)	4 (4.2)	0.352
- No	1044 (91.7)	27 (2.6)	
ASA-score			
- 1	318 (27.9)	1 (0.6)	0.003
- 2	549 (48.2)	17 (3.1)	
- 3	252 (22.1)	10 (4.0)	
- 4	20 (1.8)	2 (10.0)	
Cholangitis			
- Yes	87 (7.6)	6 (6.9)	
- No	1053 (92.4)	25 (2.4)	0.013
Previous sphincterotomy			
- Yes	477 (41.9)	14 (2.9)	0.710
- No	661 (58.1)	17 (2.6)	
Cannulation			
- Yes	1053 (92.4)	2 (2.3)	0.802
- No	87 (7.6)	29 (2.8)	
Difficult cannulation			
- Yes	278 (24.4)	7 (2.5)	0.890
- No	861 (75.6)	23 (2.6)	
Double wire			
- Yes	92 (8.1)	1 (1.1)	0.315
- No	1048 (91.9)	30 (2.9)	
Septotomy			
- Yes	32 (2.8)	1 (3.1)	0.886
- No	1108 (97.2)	30 (2.7)	
EUS + ERCP			
- Yes	107 (9.4)	6 (5.6)	0.054
- No	1032 (90.6)	25 (2.4)	

Abbreviations: N, number; EUS + ERCP, execution of endoscopic ultrasound (EUS) and endoscopic retrograde cholangiopancreatography (ERCP) during the same anesthesiologist-administered sedation session.

**Table 3 medicina-61-02172-t003:** Adjusted analysis on the risk of post-ERCP aspiration pneumonia. Adjusted OR is calculated using the Firth logistic regression model.

	Adjusted pEP OR (95% CI)	*p*-Value
Age	0.99 (0.96–1.02)	0.491
Sex (female)	1.08 (0.52–2.3)	0.839
Malignant stricture	1.03 (0.47–2.25)	0.947
ASA-score	2.09 (1.24–3.50)	0.005
Cholangitis	3.47 (1.34–9.01)	0.011
EUS + ERCP	3.55 (1.33–9.46)	0.011
Non-difficult cannulation	1.18 (0.49–2.83)	0.705

Abbreviations: OR, Odds Ratio; CI, Confidence Interval; ASA-score, American Society of Anesthesiologists score; EUS + ERCP, execution of endoscopic ultrasound (EUS) and Endoscopic retrograde cholangiopancreatography (ERCP) during the same anesthesiologist-administered sedation session.

**Table 4 medicina-61-02172-t004:** Adjusted analysis on the risk of 30-day mortality and length of hospital stay. The 30-day mortality-adjusted OR was calculated using the Firth logistic regression model; the hospital LOS-adjusted IRR was calculated using the negative binomial regression model.

	30-Day Mortality-Adjusted OR (95% Conf. Interval)	*p*-Value	Hospital LOS-Adjusted IRR (95% Conf. Interval)	*p*-Value
Age	1.03 (0.99–1.07)	0.160	1.00 (1.00–1.00)	0.648
Sex (female)	2.43 (1.09–5.40)	0.030	0.83 (0.76–0.91)	<0.001
pEP	5.24 (1.53–17.98)	0.008	2.42 (1.86–3.14)	<0.001
Malignant stricture	7.54 (2.35–24.16)	0.001	1.28 (1.16–1.40)	<0.001
ASA-score	1.97 (1.10–3.51)	0.022	1.15 (1.08–1.23)	<0.001
Non-difficult cannulation	0.72 (0.32–1.58)	0.409	0.68 (0.62–0.76)	<0.001

Abbreviations: pEP, post-ERCP aspiration pneumonia; OR, Odds Ratio; ASA-score, American Society of Anesthesiologists score; LOS, length of stay.

## Data Availability

The data presented in this study are available on request from the corresponding author.

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
