# Peer review of "Aspiration Pneumonia After ERCP Under Anesthesiologist-Administered Sedation: Prevalence, Risk Factors and Clinical Outcomes of an Underestimated Adverse Event"

_medicina, 2025, doi:10.3390/medicina61122172_

Round 1

Reviewer 1 Report

Comments and Suggestions for Authors

This is a well-conducted and clinically relevant study addressing an important but underexplored topic — aspiration pneumonia following ERCP performed under anesthesiologist-administered sedation. The manuscript provides valuable insights into the prevalence, risk factors, and outcomes of this complication in a large cohort of patients. The work is clearly written. The findings have clear clinical implications for both endoscopists and anesthesiologists.

The study focuses on an overlooked adverse event in ERCP, adding novel and clinically significant data to the literature. More than 1,100 consecutive ERCP procedures analyzed, strengthening the reliability of the findings. Identification of key independent risk factors (ASA score, pre-ERCP cholangitis, combined EUS+ERCP session) provides actionable information for clinical practice.

My suggestions:

The manuscript would benefit from minor English editing and correction of typographical errors (e.g., “perfomred”, “Unifficult”).

Ensure consistent use of abbreviations (pEP, EUS+ERCP) throughout the text and tables.

Please clarify how post-ERCP aspiration pneumonia was distinguished from other causes of post-procedural fever (e.g., cholangitis, nosocomial infections).

It would be informative to specify whether the 30-day deaths were directly related to aspiration pneumonia or to underlying disease (e.g., malignancy).

The discussion is well organized, but the authors could briefly expand on potential preventive strategies — e.g., selective intubation for high-risk patients or limiting combined EUS+ERCP sessions.

A short comment on the possible influence of prophylactic antibiotics on pEP incidence could also be valuable.

Some references appear not formatted according to the journal’s guidelines; please revise accordingly.

Comments on the Quality of English Language

The manuscript would benefit from minor English editing and correction of typographical errors (e.g., “perfomred”, “Unifficult”).

Author Response

This is a well-conducted and clinically relevant study addressing an important but underexplored topic — aspiration pneumonia following ERCP performed under anesthesiologist-administered sedation. The manuscript provides valuable insights into the prevalence, risk factors, and outcomes of this complication in a large cohort of patients. The work is clearly written. The findings have clear clinical implications for both endoscopists and anesthesiologists.

The study focuses on an overlooked adverse event in ERCP, adding novel and clinically significant data to the literature. More than 1,100 consecutive ERCP procedures analyzed, strengthening the reliability of the findings. Identification of key independent risk factors (ASA score, pre-ERCP cholangitis, combined EUS+ERCP session) provides actionable information for clinical practice.

Answer: Thank you very much for this favorable comment.

My suggestions:

  1. The manuscript would benefit from minor English editing and correction of typographical errors (e.g., “perfomred”, “Unifficult”).

Answer: Thank you very much. Minor English editing has been performed.

  1. Ensure consistent use of abbreviations (pEP, EUS+ERCP) throughout the text and tables.

Answer: Thank you very much for the comment. All the abbreviations have been double-checked.

  1. Please clarify how post-ERCP aspiration pneumonia was distinguished from other causes of post-procedural fever (e.g., cholangitis, nosocomial infections).

Answer: Thank you very much for this important comment. As reported in the Methodology Section, post-ERCP pneumonia diagnosis was considered in case of new-onset fever (>37.5°C) and/or cough arising within 48 hours after ERCP. Additionally, the performance of X-ray showing a new-onset radiological evidence of pneumonia was needed for the diagnosis. This strict inclusion criterion was used to ensure certainty of the diagnosis. However, it may have led to an underestimation of the problem.

  1. It would be informative to specify whether the 30-day deaths were directly related to aspiration pneumonia or to underlying disease (e.g., malignancy).

Answer: Thank you very much for this important comment. Unfortunately, when multiple complications occur, it becomes difficult to precisely determine the cause of death. However, the association of pneumonia with 30-day mortality at adjusted analysis confirms its important role. Among the 28 patients who died, 5 died exclusively for the underlying malignant disease and 1 for retroperitoneal perforation. The remaining patients died of sepsis (cholangitis and post-ERCP pneumonia) or multiorgan failure related to their complex clinical situation. Therefore, clearly attributing death just to pneumonia is extremely difficult in such a complex clinical scenario.

  1. The discussion is well organized, but the authors could briefly expand on potential preventive strategies — e.g., selective intubation for high-risk patients or limiting combined EUS+ERCP sessions. A short comment on the possible influence of prophylactic antibiotics on pEP incidence could also be valuable.

Answer: Thank you very much for this important comment. We fully agree with the reviewer. However, we have no strong data suggesting that limiting the use of ERCP and EUS in the same session could be beneficial because we don't know the risk of aspiration pneumonia in those undergoing two ERCP and EUS in separate sessions. This would certainly be investigated in future prospective studies. Finally, to better clarify the role of possible preventive strategies we have modified the paragraph as follows:Our data might suggest that in patients with ASA score >2, cholangitis or scheduled EUS and ERCP in the same sedation session (or more than 1 of these factors), preventive strategies might be considered to reduce the risk of pEP. No clear and definitive medical or anesthesiological strategies have been identified to avoid aspiration pneumonia. Oro-tracheal intubation and early post-procedure chest X-Ray with possible initiation of antibiotic therapy might be considered and investigated.”

  1. Some references appear not formatted according to the journal’s guidelines; please revise accordingly.

Answer: Thank you very much. References were formatted according to guidelines.

Reviewer 2 Report

Comments and Suggestions for Authors

Comments:

1. The introduction would be stronger if it were more concise and purpose-driven. Consider tightening the background to highlight only the essential context leading to your study question. The final paragraph should clearly state the specific knowledge gap—namely, the lack of focused research on post-ERCP pancreatitis under anesthesiologist-administered sedation—followed by a clear hypothesis and the primary and secondary outcomes. Emphasizing the novelty of evaluating same-session EUS and ERCP as a distinct factor would help readers immediately grasp what sets this study apart and why it fills an important gap in the existing literature.

2. Given the mechanistic link to aspiration, please report for each case (or at least by group): agents (propofol/opioid/benzodiazepine/ketamine/dexmedetomidine), doses, depth targets (Ramsay/BIS), oxygen modality (standard vs high-flow), airway maneuvers (oropharyngeal airway, gastric suction), and rescue events.

3. Patients selected for same-session procedures may have more complex malignant obstruction, biliary sepsis, or difficult access—factors that themselves predispose to aspiration. Please describe selection criteria for same-session scheduling and, if possible, adjust for procedure duration, indication granularity (e.g., distal vs hilar malignancy), and therapy performed (sphincterotomy, dilation, stent type).

4. Mandatory pre-ERCP CXR is unusual and may introduce selection (and diagnostic) biases. Describe how pre-existing atelectasis or chronic changes were handled.

5. Clinical covariates likely relevant to aspiration risk were not modeled.

6. Please report and, where feasible, adjust for: BMI/obesity, OSA, GERD/hiatal hernia, prior stroke/neuromuscular disease, narcotic use, gastric outlet obstruction, NPO interval/last oral intake, esophageal motility disorders, and emergent vs elective timing.

7. Female sex associates with lower LOS (IRR 0.83) but higher mortality (OR 2.43). Provide a clinical rationale. 

8. The Discussion suggests “antibiotic prophylaxis and/or intubation” for high-risk patients. Please nuance: antibiotics do not prevent aspiration (they may treat it) and should not be routine without infectious indications.

9. Replace “Undifficult cannulation” with “Non-difficult cannulation” or treat “difficult cannulation” as the binary variable (Yes/No) consistently across tables and models.

10. Ensure consistent naming of outcomes (“pEP,” “30-day mortality,” “LOS”) and consistent numerators/denominators across text and tables. 

Author Response

  1. The introduction would be stronger if it were more concise and purpose-driven. Consider tightening the background to highlight only the essential context leading to your study question. The final paragraph should clearly state the specific knowledge gap—namely, the lack of focused research on post-ERCP pancreatitis under anesthesiologist-administered sedation—followed by a clear hypothesis and the primary and secondary outcomes. Emphasizing the novelty of evaluating same-session EUS and ERCP as a distinct factor would help readers immediately grasp what sets this study apart and why it fills an important gap in the existing literature.

Answer: Thank you very much for this comment. As suggested the introduction has been shortened. Moreover, the final paragraph was modified as follows to make knowledge gap, primary and secondary outcomes more clear: “Definitive data on prevalence and risk factors of pEP is lacking. The identification of patient-related and ERCP-related risk factors for pEP may be useful in clinical practice to select which patient should undergo orotracheal intubation. The main aim of this study was to investigate the incidence of pEP in a tertiary academic centre. Our secondary aims were the identification of potential risk factors for pEP and investigating the impact of pEP on 30-day mortality and hospital length of stay (LOS).

  1. Given the mechanistic link to aspiration, please report for each case (or at least by group): agents (propofol/opioid/benzodiazepine/ketamine/dexmedetomidine), doses, depth targets (Ramsay/BIS), oxygen modality (standard vs high-flow), airway maneuvers (oropharyngeal airway, gastric suction), and rescue events.

Answer: Dear Reviewer, we thank you for this remark. Due to the retrospective nature of this study, it is not possible to describe each patient’s procedure. Nevertheless, we can provide the information related to the internal protocol that is daily applied within our institution. In our clinical practice, each endoscopic activity is evaluated as for the anesthesiological procedure, taking into account both the patient’s characteristics and the in fieri procedure. General anesthesia and endotracheal intubation are chosen, independently from patient’s status and conditions, if procedures are expected to last more than 45 minutes (e.g. spy glass use), or if high quantity og gastric lavage are forseen (e.g. entero-gastro anastomosis), and/or if liquid or semi-liquid material within the oesophageal-gastric-duodenal lumen is expected (e.g. hematemesis). General anesthesia and endotracheal intubation are also chosen for emergency procedures in critically ill patients and for patients scored ASA ≥ 3.

When general anesthesia and endotracheal intubation are not forseen, all patients receive anxiolysis (midazolam, 1 mg iv), light opiod analgesia (fentanyl, 50-100 mcg iv), and undergo the procedure using Marsh or Schnider target-controlled propofol infusion models (according to anesthesiologist’s preference), aiming at Richmod Agitation-Sedation (RASS) Scale of -2/-3. When deeper sedation plans are obtained, advanced airway management is always required and applied. To make this point more clear the following sentence was added to the Methodology Section: “Patients receive anxiolysis (midazolam, 1 mg iv), light opiod analgesia (fentanyl, 50-100 mcg iv), and undergo the procedure using Marsh or Schnider target-controlled propofol infusion models (according to anesthesiologist’s preference), aiming at Richmod Agitation-Sedation (RASS) Scale of -2/-3. When deeper sedation plans are obtained, advanced airway management is always required and applied.

  1. Patients selected for same-session procedures may have more complex malignant obstruction, biliary sepsis, or difficult access—factors that themselves predispose to aspiration. Please describe selection criteria for same-session scheduling and, if possible, adjust for procedure duration, indication granularity (e.g., distal vs hilar malignancy), and therapy performed (sphincterotomy, dilation, stent type).

Answer: Thank you very much for this important comment. We apologize for the lack of clarity. At our center, the possibility of scheduling EUS and ERCP in the same anesthesia session does not depend on clinical reasons but only on organizational reasons. Therefore, a combined EUS-ERCP procedure is only performed if a free double slot in the endoscopy schedule is available, otherwise the scheduling of the two procedures is separate. To make this point more clear, the following sentences were added in the Methodology Section: According to the internal organization of the Endoscopy Unit, EUS and ERCP can be performed in the same anaesthesia session only if a free double slot in the endoscopy schedule is available. Otherwise, the scheduling of EUS and ERCP is separate.

  1. Mandatory pre-ERCP CXR is unusual and may introduce selection (and diagnostic) biases. Describe how pre-existing atelectasis or chronic changes were handled.

Answer: We thank the Reviewer for this remark. When atelectasis or chronic pleuro-pulmonary changes are detected before the procedure, our behaviour varies according to the procedure urgency. If the procedure can be postpone, we will ask the medical team in charge for patient’s care to modify/implement his/her medical and operative treatment so as to improve patient’s conditions. If the procedure cannot be postponed or if no improvements could be recorded after treatment attempts, the patient will be scheduled for general anesthesia/endotracheal intubation and a post-operative care unit (PACU) admission will be forseen.

  1. Clinical covariates likely relevant to aspiration risk were not modeled. Please report and, where feasible, adjust for: BMI/obesity, OSA, GERD/hiatal hernia, prior stroke/neuromuscular disease, narcotic use, gastric outlet obstruction, NPO interval/last oral intake, esophageal motility disorders, and emergent vs elective timing.

Answer: Thank you very much for your comment. We fully agree that adding additional variables would make the analysis more complete. However, since this is a retrospective study, some of these are not accessible and therefore were not analyzed. This is an explorative study aimed at investigating the actual prevalence of the problem. Future prospective studies will be organized taking into account all these possible variables. However, OSA was considered as part of respiratory comorbidities. NPO interval is routinely >8 hours and the following sentence was added in the Methodology Section to make this point more clear: “Fasting for at least 8 hours before the procedure is required”.

Finally, we considered all the procedures performed for cholangitis as emergent and all the procedures performed for other reasons as elective.

  1. Female sex associates with lower LOS (IRR 0.83) but higher mortality (OR 2.43). Provide a clinical rationale. 

Answer: Thank you very much for this important comment. This data is not easily interpretable. However, other studies have shown a possible higher risk of mortality in women, especially in cases of infectious diseases and sepsis. For example, the paper of Pietropaoli et al. (Pietropaoli AP, Glance LG, Oakes D, Fisher SG. Gender differences in mortality in patients with severe sepsis or septic shock. Gend Med. 2010 Oct;7(5):422-37.) including 18.757 ICU patients showed that, although the incidence of sepsis is higher in men than in women, females with severe sepsis/septic shock had a higher risk of dying in the hospital compared to males. This difference remained even after multivariable adjustment. Additionally, the authors showed in their large study that hospital stay of length was significantly shorter in women even when excluding hospital non-survivors. The authors concluded that further investigation is required to determine the reason for this difference. Similar data have been published even for other diseases such as stroke (Abdel-Fattah AR, Pana TA, Smith TO, Pasdar Z, Aslam M, Mamas MA, Myint PK. Gender differences in mortality of hospitalised stroke patients. Systematic review and meta-analysis. Clin Neurol Neurosurg. 2022 Sep;220:107359).

  1. The Discussion suggests “antibiotic prophylaxis and/or intubation” for high-risk patients. Please nuance: antibiotics do not prevent aspiration (they may treat it) and should not be routine without infectious indications.

Answer: Thank you very much for this important comment. We fully agree and the sentence was modified as follows: No clear and definitive medical or anesthesiological strategies have been identified to avoid aspiration pneumonia. Oro-tracheal intubation and early post-procedure chest X-Ray with possible initiation of antibiotic therapy might be considered and investigated.

  1. Replace “Undifficult cannulation” with “Non-difficult cannulation” or treat “difficult cannulation” as the binary variable (Yes/No) consistently across tables and models.

Answer: Undifficult cannulation has been replaced with non-difficult cannulation.

  1. Ensure consistent naming of outcomes (“pEP,” “30-day mortality,” “LOS”) and consistent numerators/denominators across text and tables. 

Answer: Thank you very much. The naming and numerators/denominators of outcomes have been double-checked.

Round 2

Reviewer 2 Report

Comments and Suggestions for Authors

The authors have addressed all of my comments, and the manuscript can be accepted for publication.

Author Response

Verona, December 1st 2025

Response to Reviewers – Manuscript Revision

Dear Editors and Reviewers,

We would like to thank you for the opportunity of submitting a revised versione of our article. We trust that our revisions adequately address your concerns and further improve the quality of our work.

Sincerly,

Nicolò de Pretis

REVIEWER

Accept after minor revision

Answer: Thank you very much for this favorable suggestion.

Comments

1. Pre-ERCP chest X-ray is “mandatory”, but the authors exclude only ONE patient. If chest X-ray is truly mandatory, the exclusion of only 1/1,182 patients indicates perfect compliance, which is improbable in real clinical settings. This may reflect: inaccurate documentation, or, incomplete reporting of exclusions. It undermines internal validity if my viewpoint is correct.

Answer: Thank you very much for this important comment. One single patient was excluded because an avialable chest X-ray was lacking. This specific patient was referred from another hospital, and a chest X-ray was performed few hours before the patient was transferred. Therefore, the X-ray was not available in the patient documentation for consultation and was excluded. For all the other patients the chest X-ray was available and visible for consultation because in our hospital the anesthesiologist does not consent to sedation without a procedural x-ray. This explains the high adherence to the pre-procedural chest X-ray performance.

2. Author state:“the included population well represent all different possible indications to ERCP”. But Table 1 shows 50% malignant strictures, which is not representative of typical ERCP cohorts. Most global cohorts have: predominantly choledocholithiasis (40–60%), far fewer malignant strictures (5–20%), etc. Thus, the population is not representative, limiting generalizability.

Answer: Thank you very much for your comment. We fully agree with the reviewer. The reason for the high prevalence of malignant biliary stricture is that the Endosocpy Unit of the University of Verona is a tertiary center for pancreatic disease and a significant proportion of patients undergoing ERCP have pancreatic cancer. Nevertheless, around 25% of the procedures were performed for biliary stones and all the main indications to ERCP are included. However, to make this point more clear, the following sentence has been added in the Discussion Section to better undreline this limitation: “finally, considering that the Endoscopy Unit of the University of Verona is a reference center for pancreatic diseases, a significant proportion of the included cases was represented by patients with malignant biliary stricture”.

3. Contradiction between unadjusted and adjusted results ignored. For example, EUS+ERCP: Unadjusted: p=0.054 (NS) and Adjusted: OR 3.55, p=0.011 (significant). The discussion treats EUS+ERCP as clearly a risk factor but does not explain the confounding effects or the shift to significance. A clear explanation is missing and necessary to be given.

Answer: Thank you very much for highlighting this important point. We agree that reporting univariable (unadjusted) results is essential for transparency and completeness of the study, as it allows readers to see the initial associations before accounting for potential confounders. However, once the multivariable (adjusted) analyses are performed and adjustments for confounding factors are applied, the unadjusted estimates become less informative and can even be misleading. In our case, the shift from a non-significant unadjusted result (p=0.054) to a significant adjusted result (OR=3.55, p=0.011) for EUS+ERCP reflects the influence of confounders such as patient clinical characteristics and procedural complexity. The adjusted model provides a more accurate estimate of the independent effect of EUS+ERCP.

4. pEP as "relatively common". The authors state: “pEP is relatively common (2.7%).” But: prior studies report 0.38–1.05% in non-intubated ERCP. Even your own data show only 31/1140 cases. Calling 2.7% “relatively common” is debatable and not internally justified. Thus pEP as having a “significant impact,” which is statistically true but clinically uncertain due to sparse events.

Answer: Thank you very much for your comment. We agree that an adverse event reported in less than 3% of the procedure is not frequent. However, a recent meta-analysis (Bishay K et al.Adverse events associated with endosocpic retrograde cholangiopancreatography: systematic review and meta-analysis. Gastroenterology 2025) reported as overall adverse events the following: pancreatitis 4.6%, bleeding 1.5%, cholangitis 2.5%, cholecystitis 0.8%, perforation 0.5%. Therefore, despite the limitations of a retrospective study, an incidence of 2.7% of post-ERCP aspiration pneumonia was considered significant by us. However, to make this point less assertive, we modified the sentence as follows: “In conclusion, pEP is a relevant adverse event of ERCP performed under anesthesiologist-administered sedation, impacting on mortality and length of hospital stay”.

5. There is NO data in the study comparing intubated vs. non-intubated ERCP and your recommended for intubation is unsupported by data.

Answer: Thank you very much for your comment. We fully agree. The hypothesis that intubation might reduce the risk of aspiration pneumonia is just speculative in this clinical setting. Future studies are needed to clarify this point. The following sentence was added to make this point less assertive: “Oro-tracheal intubation and early post-procedure chest X-ray with possible initiation of antibiotic therapy might be considered and investigated”.

6. ASA score, cholangitis, etc are associations and not causations - discussion treats associations as mechanistic explanations, which is over-interpretation.

Answer: Thank you very much for this important comment. We agree with the reviewer that a clear cause-effect has not been identified. However, we tried to hypothesize possible explanations that could justify the increased risk of aspiration pneumonia in some subgroups. Future studies are required to deeper investigate these associations.

7. Without economic analysis you claim about cost benefits which is unwarranted.

Answer: Thank you very much for your comment. The sentence was modified as follows: Therefore, the identification of risk factors for pEP and the development of specific preventive strategies might have a relevant clinical impact in patient management.